# The Adaptor Protein UvSte50 Governs Fungal Pathogenicity of *Ustilaginoidea virens* via the MAPK Signaling Pathway

**DOI:** 10.3390/jof8090954

**Published:** 2022-09-11

**Authors:** Huijuan Cao, Hao Gong, Tianqiao Song, Mina Yu, Xiayan Pan, Junjie Yu, Zhongqiang Qi, Yan Du, Yongfeng Liu

**Affiliations:** 1Institute of Plant Protection, Jiangsu Academy of Agricultural Sciences, Nanjing 210014, China; 2College of Plant Protection, Nanjing Agricultural University, Nanjing 210095, China

**Keywords:** rice false smut, *Ustilaginoidea virens*, adaptor protein UvSte50, MAPK signaling pathways, pathogenicity

## Abstract

The mitogen-activated protein kinase (MAPK) signaling pathways regulate diverse cellular processes and have been partially characterized in the rice false smut fungus *Ustilaginoidea virens*. UvSte50 has been identified as a homolog to *Saccharomyces cerevisiae* Ste50, which is known to be an adaptor protein for MAPK cascades. Δ*Uvste50* was found to be defective in conidiation, sensitive to hyperosmotic and oxidative stresses, and non-pathogenic. The mycelial expansion of Δ*Uvste50* inside spikelets of rice terminated at stamen filaments, eventually resulting in a lack of formation of false smut balls on spikelets. We determined that UvSte50 directly interacts with both UvSte7 (MAPK kinase; MEK) and UvSte11 (MAPK kinase kinase; MEKK), where the Ras-association (RA) domain of UvSte50 is indispensable for its interaction with UvSte7. UvSte50 also interacts with UvHog1, a MAP kinase of the Hog1-MAPK pathway, which is known to have important roles in hyphal growth and stress responses in *U. virens*. In addition, affinity capture–mass spectrometry analysis and yeast two-hybrid assay were conducted, through which we identified the interactions of UvSte50 with UvRas2, UvAc1 (adenylate cyclase), and UvCap1 (cyclase-associated protein), key components of the Ras/cAMP signaling pathway in *U. virens*. Together, UvSte50 functions as an adaptor protein interacting with multiple components of the MAPK and Ras/cAMP signaling pathways, thus playing critical role in plant infection by *U. virens*.

## 1. Introduction

The ascomycete fungus *Ustilaginoidea virens* (teleomorph: *Villosiclava virens*) is the causal agent of rice false smut disease, which infects rice floral organs and develops false smut balls in the panicles [1,2]. This disease has recently become one of the most severe grain diseases in rice-growing areas [3]. Besides causing yield losses, *U. virens* seriously threatens food security, as its mycotoxins are harmful to the nervous systems of animals, like *Fusarium* mycotoxins and aflatoxins [4,5,6]. As a biotrophic parasitic pathogenic fungus, the pathogenic mechanism of *U. virens* has a certain specificity. The conidia germinate on the outer surface of the glume, after which the hyphae enter the inside of the glume through the gap between the inner and outer glumes. The hyphae initially infect the stamen filaments, then extend to the stigma, anther, and other flower organs. Afterwards, the mycelium proliferates, wrapping the entire flower organ and filling the interior of the grain [2]. *U. virens* competes with rice pollen fertilization, hijacking rice nutrients for mycelial growth by simulating ovary fertilization [2,7]. During the infection process, the germ hyphae of *U. virens* cannot differentiate into specialized infection structures, such as appressoria or haustoria. Furthermore, the hyphae only grow in the intercellular space between host cells, and does not penetrate the plant cell wall and enter the interior space of host cells [1]. At present, the underlying infection mechanisms of *U. virens* are not well-understood.

The mitogen-activated protein kinase (MAPK) signaling pathways are known to regulate morphogenesis and pathogenesis in numerous fungal pathogens [8,9,10]. The MAPK modules are usually composed of three protein kinases: MAPK kinase kinase (MAPKKK or MEKK), MAPK kinase (MAPKK or MEK), and MAPK. During signal transduction, the sequential activation of the three interconnected protein kinases leads to the activation of downstream transcription factors and the expression of specific genes in response to extracellular stimuli [11,12,13,14]. At least five MAPK pathways have been characterized in *Saccharomyces cerevisiae*, which respectively regulate the pheromone response (Fus3/Kss1), filamentous growth (Kss1), cell wall integrity (Slt2), high osmolarity regulation (Hog1), and spore wall assembly (Smk1) [15]. In phytopathogenic fungi, there are at least three MAPKs, which are homologous to yeast Hog1, Slt2, or Fus3/Kss1 [16]. UvPmk1 (homologous to yeast Fus3/Kss1) of *U. virens* is essential for conidiation, stress response, and pathogenicity [17]. UvSlt2 has a conserved role in cell wall integrity, and deletion mutants have been shown to lose virulence [18,19]. UvHog1 regulates hyphal growth, stress responses, and secondary metabolism, but the regulation of pathogenicity has not been mentioned in the article reporting its characterization [20]. A novel transcription factor, *UvCGBP1*, functions through the MAPK pathway to regulate the development and virulence of *U. virens* [19]. Although the key MAP kinase of each MAPK pathway has been characterized in *U. virens*, the regulatory mechanism and potential adaptor proteins between the MAPK pathways have not yet been well-characterized.

In the process of cellular signal transduction, some receptors need to be mediated by adapter proteins in order to receive and transmit signals to produce relevant effects in response. In *S. cerevisiae*, Ste50 is used as an adaptor protein to couple the Cdc42–Ste20 complex to Ste11 (MEKK), which then participates in the Fus1/Fus3 and Hog1 MAPK pathways, regulating pheromone response, filamentous growth, and resistance to hyperosmotic stress [21]. Ste50 is also associated with the Ras/cAMP signaling pathway, possibly due to the presence of the RA domain, which can interact with Ras1 and Ras2 [21,22]. In *Cryptococcus neoformans*, Ste50 is involved in pheromone sensing and sexual reproduction through the Cpk1 MAPK pathway, but not in stress responses and virulence factor production, controlled by HOG and Ras/cAMP signaling pathways [23]. In *Fusarium graminearum*, FgSte50 interacts with FgSho1, FgSte7, and FgSte11 to affect the activities of MAPK signaling pathways, regulating the pathogenicity of *F. graminearum* [24,25]. In *Magnaporthe oryzae*, Mst50 (homologous to yeast Ste50) is used as a backbone protein to stabilize the binding of Mst11 (homologous to yeast Ste11) and Mst7 (homologous to yeast Ste7) to activate the Pmk1(homologous to yeast Fus3/Kss1) gene, thus regulating the formation of appressoria and pathogenicity [26]. In *Ustilago maydis*, Ubc2 (homologous to yeast Ste50) contains C-terminal SH3 domains required for pathogenicity that typically bind Ste11 orthologues and Ras, but not the Ste7 (MAEK) or Fus3 (MAPK) components [27]. In *Botrytis cinerea*, Ste50 is involved in the regulation of appressorium formation and infection hyphae growth [28]; however, the functions of UvSte50 in the development and infection processes of *U. virens* remain unclear.

In this study, we functionally characterize the *UvSTE50* gene in the *U. virens* Pmk1-, Hog1-, and Ras/cAMP signaling pathways. The Δ*Uvste50* mutant is found to be defective in conidiation, sensitive to hyperosmotic and oxidative stresses, and is non-pathogenic on rice panicles. UvSte50 is found to directly interact with UvSte7, UvSte11, UvHog1, UvRas2, UvAc1, and UvCap1 through affinity capture–mass spectrometry analysis and yeast two-hybrid assay. These results demonstrate that UvSte50 is involved in multiple signaling pathways which control development and plant infection processes in *U. virens*.

## 2. Materials and Methods

### 2.1. Strains and Growth Conditions

The *U. virens* strain Jt209 [29] was used as the wild-type strain for subsequent strain construction. The wild-type strain and all transformants generated in this study were routinely cultured on potato sucrose agar (PSA: 200 g/L potato, 20 g/L sucrose, and 15 g/L agar) plates at 28 °C [30]. *Agrobacterium tumefaciens* strain AGL1 was used for T-DNA insertional transformation of *U. virens* [30]. *Escherichia coli* strain DH-5α was used for construction of various plasmids [30].

### 2.2. Construction of Gene Deletion and Complementation Mutants

Gene deletion and complementation vectors were constructed and then transformed into *U. virens* as described previously [31]. The gene replacement vector pMD19-*UvSTE50* was constructed by inserting the hygromycin gene (*HYG*) between the two flanking sequences of *UvSTE50*, which were amplified with primers 1F/2R and 3F/4R (listed in Appendix A). The CRISPR-Cas9 vectors gRNA-*UvSTE50* were constructed by cloning double-stranded gRNA spacers to the BsmBI sites of pmCas9: tRp-gRNA [18,32]. The resulting vectors pMD19-*UvSTE50* and gRNA-*UvSTE50* were co-transformed into protoplasts of the wild-type strain Jt209 by PEG-mediated transformation, as previously described [31]. Hygromycin-resistant transformants were screened by PCR assays with primers 5F/6R, 7F/8R, and 9F/10R (Appendix A). Putative *UvSTE50* gene deletion mutants were further verified by sequencing.

The complementation vector pKO1-*UvSTE50* was constructed by cloning a 3.7 kb fragment of *UvSTE50* amplified with primers *UvSTE50*-comF/comR (Appendix A) into the G-418 resistance vector pKO1-NEO. Then, the resulting *UvSTE50* complementation vector was transformed into ∆*Uvste50* using the *Agrobacterium*-mediated transformation (ATMT) method [33]. G-418 resistance transformants were screened by RT-PCR at the mRNA level (Appendix A).

### 2.3. Analysis of Mycelial Growth and Conidiation

For mycelial growth assays, each *U. virens* strain was incubated for 20 days in the dark at 28 °C in PSA medium, after which the colony diameters were measured. Conidial development was assessed by harvesting conidia from 6-day-old liquid PSB cultures. The conidial morphology was observed under a microscope, and the concentration of conidial suspension was recorded using a hemocytometer [17]. For conidial germination assays, conidia were diluted to a concentration of 1 × 10^6^ conidia/mL with PSB, then incubated at 28 °C with shaking (150 rpm). Conidial germination structures were observed 20 h post-incubation [34]. All experiments were performed three times with three replicates.

### 2.4. Virulence and Plant Infection Assays

Pathogenicity and plant infection assays of each *U. virens* strain on a susceptible rice cultivar (Liangyoupeijiu) were assessed as described previously [30]. The *U. virens* strains were cultured for 6–7 days at 28 °C in PSB medium with shaking at 150 rpm, then homogenized with a tissue blender (Waring Commercial Blender 8011S, USA). A mixture of mycelia and conidia were diluted to a concentration of 1 × 10^6^ conidia/mL with PSB, and 1–2 mL of mycelia and conidia suspensions were injected into the panicles before rice heading stage using sterilized syringes. The inoculated rice plants were cultivated in a humid environment, and the number of rice false smut balls per panicle was counted at 30 dpi [31]. The expansion of infection hyphae inside the spikelets of Jt209, Δ*Uvste50*, and *Uvste50-c* was observed at 3 dpi, 5 dpi, 10 dpi, and 14 dpi. The plant infection assays were repeated three times.

### 2.5. Stress Adaptation Assays

To assess the effect of UvSte50 in regulating *U. virens* adaptation to pathogenesis- or morphogenesis-associated stress, the radial growth of Jt209, Δ*Uvste50*, and *Uvste50-c* was compared on PSA plates containing hyperosmotic stress agents (NaCl or sorbitol), cell-wall-damaging agents (calcofluor white, CFW; sodium dodecyl sulfate, SDS; or Congo red, CR), or an oxidative stress agent (H_2_O_2_). For this purpose, 5 mm mycelial plugs of each strain were inoculated in PSA plates containing exogenous 0.5 M NaCl, 0.6 M sorbitol, 500 µg/mL CFW, 100 µg/mL CR, 0.05% SDS, or 0.05% H_2_O_2_. The plates were incubated at 28 °C for 20 days in the dark, and colony diameters and inhibition rates were calculated as described previously [31,35,36]. To determine the influence of hyperosmotic and oxidative stress during conidium germination, both the conidia of mutant and wild type were incubated in liquid PSB media or with 0.3 M NaCl, 0.3 M sorbitol, or 0.01% H_2_O_2_, and germ tubes were observed after 20 h [20]. Each treatment was repeated three times.

### 2.6. RT-qPCR and Transcriptome Sequencing Analysis

Gene expression was evaluated by RT-qPCR using specific primers (listed in Appendix A). To detect the transcript level of *UvSTE50*, samples of inoculated rice spikelets were collected at 0, 1, 2, 3, 5, 7, 10, 14, and 30 days post-inoculation. Total RNA was extracted using an RNA extraction kit (Biotech, Beijing, China) and cDNA was synthesized with a PrimeScriptTM RT reagent kit with gDNA Eraser (TaKaRa, Osaka, Japan) [36]. qPCR was performed using TB Green Premix Ex TaqTM II (Tli RNaseH Plus) (TaKaRa, Osaka, Japan), and detected on an ABI Q6 Real-Time System.

Total RNA of the wild-type Jt209 and Δ*Uvste50* mutant were extracted from 4-day-old vegetative hyphae using an RNA extraction kit (Biotech, Beijing, China), and RNA integrity was confirmed using an RNA Nano 6000 Assay Kit and a Bioanalyzer 2100 system (Agilent Technologies, CA, USA). RNA-seq library preparation and transcriptome sequencing using an Illumina Novaseq platform were performed at Novogene biomedical technology Co., Ltd. (Beijing, China). Clean reads form RNA-seq data of each sample were aligned to a *U. virens* reference genome (GenBank assembly accession: GCA_000687475.2) [37]. Differential expression analysis of Jt209 and Δ*Uvste50* was performed using the DESeq2 R package (1.20.0), where log_2_(fold change) ≥ 1 and p_adj_ ≤ 0.05 were selected as the DEGs screening criteria. Gene Ontology (GO) enrichment analysis of differentially expressed genes was implemented through the cluster Profiler R package, in which gene length bias was corrected.

### 2.7. Generation of Green Fluorescent Protein (GFP) Fusion Cassettes

To construct the Uvste50–GFP fusion cassette, *UvSTE50* ORF was amplified with the primer pair *UvSTE50*–GFPF/GFPR (Appendix A). The resulting PCR product was cloned into the BamHI and SmaI sites of vector pKD1-GFP [31], then transformed into the *UvSTE50* deletion mutant by ATMT, as described previously [33]. Hygromycin-resistant transformants were observed for GFP signals with a Zeiss LSM780 confocal microscope (Carl Zeiss AG, Jena, Germany).

### 2.8. Yeast Two-Hybrid Assay

Protein–protein interactions were identified using the Matchmaker^®^ Gold Yeast Two-hybrid System (Clontech, LA, USA), according to the user manual. The coding sequences of each tested gene were amplified from the Jt209 cDNA with primer pairs (listed in Appendix A) and inserted into vector pGBKT7 or pGADT7, respectively. The pairs of yeast two-hybrid plasmids were co-transformed into *S. cerevisiae* strain Y2HGold following the yeast transformation protocol. In addition, plasmids pGBKT7-53 and pGADT7 were used as positive controls, while plasmids pGBKT7-Lam and pGADT7 served as negative controls. Transformants were grown on SD/-Leu/-Trp and SD/-Ade/-His/-Leu/-Trp media for 3–5 days at 30 °C, in order to assess binding activity. Three independent experiments were performed to confirm the results.

### 2.9. Affinity Capture–Mass Spectrometry Analysis

The empty pKD1-GFP and *UvSTE50*-GFP fusion was transformed into the *UvSTE50* deletion mutant by ATMT, in order to obtain positive control and experimental groups, respectively. Total proteins of the resulting transformants were prepared from mycelia as described previously [18]. Then, 50 μL of GFP-trap agarose (ChromoTek, Planegg, Germany) was added to capture the UvSte50-GFP interaction proteins, according to the manufacturer’s protocols. After incubation at 4 °C for 1–2 h, the agarose was washed three times with 500 μL of washing buffer (10 mM Tris-HCl, pH 7.5; 150 mM NaCl; 0.5 mM EDTA; 0.05% Nonidet TM P40 Substitute). Proteins bound to the beads were boiled at 95 °C for 5 min with 60 μL 2 × SDS-sample buffer. After centrifugation at 2500× *g* for 2 min at 4 °C, proteins in the supernatant were analyzed by SDS-PAGE/Western blot. Then, the elution proteins were analyzed by mass spectrometry (MS) at BGI biomedical technology Co., Ltd. (Shenzhen, China).

## 3. Results

### 3.1. Identification of Ste50 Ortholog in U. Virens

Using *Saccharomyces cerevisiae* adaptor protein Ste50 as a query, we identified only one *STE50* ortholog, UV8b_07862 (QUC23621.1, hereafter UvSte50) from the *U. virens* genome (GenBank assembly accession: GCA_000687475.2) with BLASTP. UvSte50 was predicted to encode a 499-amino-acid protein with 41.38% identity to the Ste50 of *S. cerevisiae*. UvSte50 contains two domains, a sterile alpha motif (SAM) domain (72–129 aa) and a Ras-association (RA) domain (379–466 aa); as shown in Figure 1A. Phylogenetic analysis of proteins indicated that UvSte50 and the yeast Ste50 were clustered into different groups, where UvSte50 shared the highest similarity with Ste50 from *Hypocrella siamensis* (Figure 1B).

### 3.2. UvSte50 Is Required for Conidiation in U. virens

To characterize the roles of UvSte50 in *U. virens*, we generated gene deletion mutants of *UvSTE50* using a homologous recombination strategy assisted with CRISPR-Cas9 system in the wild-type strain Jt209 [18]. Deletion strains were selected and further confirmed by RT-PCR and sequencing analysis (Appendix A). The mutant Δ*Uvste50* grew similarly to the wild-type Jt209 in PSA plates (Figure 2A), while Δ*Uvste50* showed a significant decrease in conidiation when inoculated in liquid PSB media (Figure 2B). Notably, the conidiation defect of Δ*Uvste50* was restored by complementation with the wild-type *UvSTE50* in the complemented strain *Uvste50-c* (Figure 2B). However, Δ*Uvste50* did not exhibit detectable changes in conidial morphology and germination (Figure 2C and Appendix A). The width and length of conidia produced by Δ*Uvste50* and the germination rate of Δ*Uvste50* showed no difference from those for the wild-type Jt209 (Appendix A). These results suggest that UvSte50 plays an important role in conidiation of *U. virens*.

### 3.3. UvSte50 Is Essential for Full Virulence in U. virens

The virulence of Δ*Uvste50* was assessed through the inoculation of mycelia and conidia suspensions into rice panicles at booting stage (5–7 days before heading). After 30 days of incubation, Δ*Uvste50* did not cause disease on grains, and no false smut balls were visible on the rice panicles (Figure 3A,B). In contrast, approximately 40–50 small false smut balls were found on each spike inoculated with wild-type Jt209 and the complemented strain *Uvste50-c* (Figure 3A,B). In order to verify a pathogenicity defect of the mutant in vivo, the GFP gene driven by the promoter of the *U. virens* histone H3 gene was introduced into Δ*Uvste50* and the wild-type Jt209, and the resulting strains Δ*Uvste50*-GFP and Jt209-GFP were used to inoculate rice panicles. As shown in Figure 3C, at 3–5 dpi, the GFP signals of Δ*Uvste50*-GFP were restricted within the stamen filaments, while GFP signals of Jt209-GFP were observed from the stamen filaments and anthers (Figure 3C). At 10–14 dpi, the spikelets infected by Jt209-GFP were full of white hyphae and the GFP signals were observed to cover floral organs, while no hyphae were observed inside the spikelets infected by Δ*Uvste50*-GFP (Figure 3D). Hyphae of *U. virens* initially infects the stamen filaments and then extends to embrace the inner floral organs inside the spikelets of rice [1,2]. Therefore, our results suggest that the infection process of Δ*Uvste50* was blocked at the stamen filaments.

The relative expression profiles of *UvSTE50* during the infection process of *U. virens* were determined by RT-qPCR. Samples of inoculated rice spikelets at 0, 1, 2, 3, 5, 7, 14, and 30 dpi were collected. The expression level of *UvSTE50* showed an upward trend in the early stage of inoculation, reaching the highest value at 5 dpi, which was 4.2-fold of the expression level at the initial inoculation (Figure 4). After that, the expression level of *UvSTE50* showed a downward trend (Figure 4). Hyphae of *U. virens* typically infect the stamen filaments at 3–5 dpi [1,2]; thus, the expression peak of *UvSTE50* at 5 dpi suggests that *UvSTE50* plays a crucial role in the early infection process of *U. virens*.

### 3.4. Culture Filtrates of ΔUvste50 Are Less Toxic to Rice Seed Germination

To investigate whether UvSte50 affects the production of phytotoxic compounds, we collected PSB culture filtrates from 5-day-old wild-type Jt209, Δ*Uvste50* mutant, and from the complemented strain *Uvste50-c*, which were subjected to rice seed germination assays. After soaking Liangyoupeijiu rice seeds in PSB medium (control group) and different culture filtrates of *U. virens* for 5 days, it was observed that the germination rate of rice seeds in each treatment was basically the same, but the germination status was quite different. In the control group treated with liquid PSB, the root length of germinated rice seeds reached 3.38 ± 0.49 cm, while the shoot length reached 1.70 ± 0.27 cm. However, the growth of roots and shoots of rice seeds treated with filtrates of *U. virens* was inhibited, to a certain extent (Figure 5A,B). The root length was only 0.41 ± 0.07 cm when rice seed were treated with filtrates of wild-type strain Jt209, while the root growth was longer when treated with filtrate of Δ*Uvste50* mutant, reaching 1.02 ± 0.18 cm (Figure 5A,B). The shoot length of rice treated with the filtrate of Jt209 was 0.60 ± 0.08 cm, while that with Δ*Uvste50* was 1.21 ± 0.20 cm (Figure 5A,B). Thus, the shoots of rice also grew longer when treated with filtrate of Δ*Uvste50* mutant, compared to those treated with the wild-type and complemented strains under the same conditions (Figure 5). These results suggest that culture filtrates of Δ*Uvste50* are less toxic, in terms of rice seed germination, and therefore, *UvSTE50* may be involved in regulating the production of phytotoxic compounds in *U. virens*.

### 3.5. UvSTE50 Is Involved in Hyperosmotic and Oxidative Regulation in U. virens

In *S. cerevisiae*, Ste50 serves as an adaptor protein in the Fus3 and HOG pathways to regulate morphogenesis-associated stress [15,22]. Therefore, we were interested in determining the sensitivity of Δ*Uvste50* to hyperosmotic, cell-wall-damaging agents, and oxidative stress. We cultured Δ*Uvste50* on PSA medium containing an exogenous hyperosmotic concentration of NaCl or sorbitol; diverse cell-wall-damaging agents, including Congo red (CR), calcofluor white (CFW), and SDS; or oxidative H_2_O_2_, in order to determine their inhibitory effects on fungal growth. On PSA medium with cell-wall-damaging agents including CR, CFW, and SDS, we failed to observe any significant difference between Δ*Uvste50* and the wild-type Jt209 (Figure 6A,B). However, the mutant Δ*Uvste50* was more sensitive to hyperosmotic and oxidative stresses. In the presence of 0.5 M NaCl or 0.6 M sorbitol, the reduction in growth rate was more severe in Δ*Uvste50* than in Jt209 or *Uvste50-c* (Figure 6A,B), indicating that UvSte50 is involved in osmoregulation in *U. virens*, possibly through cross-talk with the HOG1 pathway [20]. We also assayed the effect of hyperosmotic stress on conidium germination. In the presence of 0.3 M NaCl, 72.0 ± 3.1% wild-type Jt209 conidia germinated after incubation for 20 h, but only 46.3 ± 4.1% of Δ*Uvste50* mutant conidia germinated (Figure 6C and Appendix A). Moreover, germ tube growth was stunted by NaCl treatment in the Δ*Uvste50* mutant (Appendix A). Similar results were obtained with germination assays on medium with 0.3 M sorbitol (Figure 6C and Appendix A).

On the PSA plates with 0.05% H_2_O_2_, Δ*Uvste50* also showed decreased tolerance, compared to Jt209 and *Uvste50-c* (Figure 6A,B). The conidium germination rate of the Δ*Uvste50* mutant was also decreased, compared to wild-type Jt209 (Figure 6C and Appendix A). Based on these results, we conclude that UvSte50 appears to be necessary for hyperosmotic and oxidative regulation in the filamentous fungus *U. virens*.

### 3.6. Deletion of UvSTE50 Affects the Expression of Genes Involved in MAPK Signaling Pathways

The genes involved in MAPK signaling pathways are known to regulate stress-related genes in *U. virens* and other fungi [16,17,18,19]. To determine whether the deletion of *UvSTE50* affected the expression of three MAPK (*UvPMK1*: UV8b_03045, *UvSLT2*: UV8b_00381 and *UvHOG1*: UV8b_04241), three MEKK (*UvSTE11*: UV8b_06470, *Uv**BCK1*: UV8b_01370 and *Uv**SSK2*: UV8b_02087), and three MEK (*UvSTE7*: UV8b_04866, *Uv**MKK2*: UV8b_02817 and *Uv**PBS2*: UV8b_02207) in *U. virens*, RNA samples were isolated from hyphae of the wild-type strain Jt209 and Δ*Uvste50* mutant, harvested from regular PSB cultures and cultures treated with 0.5 M NaCl or 0.05% H_2_O_2_ for 5 h. In the wild-type strain Jt209, the expression of all tested genes presented no significant change when treated with 0.05% H_2_O_2_, while NaCl treatment resulted in increased expression of *UvPMK1*, *UvSLT2*, and *UvSTE11* (Figure 7). In the Δ*Uvste50* mutant, the expression of *UvPMK1* and *UvSLT2* was up-regulated by over two-fold in the presence of NaCl, and the expression of *UvHOG1* and *UvSSK2* was reduced by over two-folds under treatment with 0.5 M NaCl or 0.05% H_2_O_2_ (Figure 7).

In comparison with the wild-type strain Jt209, the expression of all nine tested genes in PSB showed no significant change in the Δ*Uvste50* mutant (Figure 7); however, compared to the wild-type, the expression of *UvHOG1*, and *UvSSK2* were reduced by over 50% in the presence of NaCl and H_2_O_2_ in the Δ*Uvste50* mutant (Figure 7). For *UvPMK1*, and *UvSTE11*, their expression in the mutant was significantly reduced when treated with H_2_O_2_ (Figure 7). However, the wild-type strain and Δ*Uvste50* mutant presented no obvious differences in expression level of *UvMMK2*, and *UvBCK1* when treated with NaCl or H_2_O_2_ (Figure 7). These results indicate that the deletion of *UvSTE50* has varying effects on these genes involved in the MAPK signaling pathways of *U. virens*.

### 3.7. Deletion of UvSTE50 Affects the Transcription of a Subset of Genes in U. virens

To further understand the regulation mechanisms of UvSte50 in *U. virens*, we compared the global gene transcription patterns of the mycelia of the Δ*Uvste50* mutant with that of the wild-type strain Jt209 through transcriptome sequencing analysis. We prepared three biological replicates for each of the mutant Δ*Uvste50* (Δ*Uvste50*_1/2/3) and wild-type Jt209 (Jt209_1/2/3). The Pearson correlation (R^2^) between samples of three wild-type (Jt209_1/2/3) or three mutant (Δ*Uvste50*_1/2/3) samples were >0.96 (Appendix A), suggesting that the transcriptome sequencing data were credible. There were 625 differentially expressed genes (DEGs) [p_adj_ ≤ 0.05, log_2_(fold change) ≥ 1] between the wild-type Jt209 and mutant Δ*Uvste50* strains (Figure 8A, Appendix A). Among the 625 DEGs, 365 genes were down-regulated and 260 genes were up-regulated in Δ*Uvste50* (Figure 8A, Appendix A).

Gene Ontology (GO) enrichment analysis was conducted to determine the functions of DEGs [38], from which all DEGs could be assigned to three functional groups: biological process, cellular component, and molecular function (Figure 8B, Appendix A). Several enriched GO terms implied that UvSte50 is required for the oxidation–reduction process (GO:0055114 and GO:0016491) and membrane (GO:0016020) in *U. virens* (Figure 8B, Appendix A).

Major facilitator superfamily (MFS) transporter genes are likely transcriptional regulated by MAPK pathways in *F. graminearum* [39]. We tested the expression of nine MFS transporter genes by RT-qPCR. Two of the nine tested genes (UV8b_06962 and UV8b_01258) were significantly up-regulated, while seven genes (UV8b_03319, UV8b_06031, UV8b_07045, UV8b_04815, UV8b_06043, UV8b_01802, and UV8b_00585) were significantly down-regulated in the mycelia of Δ*Uvste50* (Figure 8C). The RT-qPCR data for these tested genes were coincident with those in the transcriptome sequencing results. Thus, MFS transporter genes are likely under transcriptional regulation by *UvSTE50* in *U. virens*. Cdtf1 is the downstream transcription factor of cAMP/PKA signaling pathways in *M. oryzae* [40], and the expression of *UvCDTF1* (UV8b_03574) was upregulated in Δ*Uvste50* (Figure 8C), while the other components of cAMP/PKA signaling pathways, including adenylate cyclase *UvAC1* [41], cAMP-dependent protein kinase *UvCPKA* and *UvCPK2* or cAMP phosphodiesterase [41], presented no significant change in expression level in the Δ*Uvste50* mutant (Appendix A).

In addition, deletion of *UvSTE50* had no significant effect on the expression levels of *STE12* (UV8b_02834), *HOX7* (UV8b_06464), *MCM1* (UV8b_04334), *SWI6* (UV8b_00240), *AP1* (UV8b_06147), and *ATF1* (UV8b_07350) orthologs that are downstream transcription factors of MAPK signaling in *S. cerevisiae* or *M. oryzae* [10,42,43,44] (Appendix A). The expression levels of genes encoding the G-proteins (Gα: UV8b_06528, UV8b_04647, and UV8b_04239; Gβ: UV8b_07865; and Gγ: UV8b_02893) were also normal in the Δ*Uvste50* mutant (Appendix A), suggesting that the expression of well-conserved upstream G-proteins, downstream transcription factors of MAPK signaling, is not affected by the deletion of *UvSTE50*.

### 3.8. UvSte50 Was Distributed as Spots in the Cytoplasm of Hyphae

To determine the sub-cellular localization of UvSte50 in *U. virens*, we generated the UvSte50-enhanced GFP fusion protein construct and transformed it into the mutant Δ*Uvste50*. G-418 resistant transformants were isolated and the resultant strain, UvSte50-GFP, was found to be fully pathogenic on rice plants. Protein gel blot analysis showed a band of expected UvSte50-GFP fusion size with anti-GFP antibody in UvSte50-GFP strain (data not shown). In the UvSte50-GFP transformant, GFP signal spots were detected in vegetative hyphae through confocal fluorescence microscopy (Figure 9A), different from the FgSte50 localized onto the cell membrane in *F. graminearum* [25] or Mst50 expressed very low in mycelia and conidia of *M. oryzae* [26]. The different sub-cellular localization may be related to functional differentiation of Ste50 in different pathogenic fungi.

### 3.9. Protein–Protein Interaction Studies to Identify UvSte50-Interacting Partners

Given that UvSte50 is indispensable for the maintenance of virulence in *U. virens*, we questioned which signaling pathway may be implicated with UvSte50. To identify UvSte50-interacting partners, we performed an affinity capture–mass spectrometry (MS) assay for UvSte50. Briefly, the total protein was isolated from the UvSte50-GFP strain, and GFP-trap agarose (ChromoTek, Planegg, Germany) was used to capture the UvSte50-interacting proteins. The elution proteins from the beads were analyzed by mass spectrometry. The Δ*Uvste50* strain transformed with empty pKD1-GFP was used as a negative control. In the assay, we discovered that UvSte7 (UV8b_04866), UvSte11 (UV8b_06470), and UvHog1 (UV8b_04241), which are homologous to yeast Ste7, Ste11, and Hog1, respectively, co-purified with UvSte50 (Table 1). We thus raised a hypothesis that UvSte50 may physically interact with the MAPK modules Ste11-Ste7-Pmk1 and UvHog1-MAPK. To test this, we conducted yeast two-hybrid assays. As shown in Figure 9B, UvSte50 interacted with UvSte7, UvSte11, and UvHog1. Although there was no interaction between UvSte50 and UvPmk1, UvSte7, and UvSte11 interacted with UvPmk1 (Figure 9B and Appendix A). The affinity capture–mass spectrometry assay also showed that cAMP signaling pathway components UvAc1 (UV8b_02467) and UvCap1 (UV8b_00969, homologous to yeast Srv2) were co-purified with UvSte50 (Table 1), and the yeast two-hybrid assay also confirmed this result (Figure 9B and Appendix A). Other proteins identified in the affinity capture mass-spectrometry assay included peroxisomal biogenesis factor 6 (UV8b_06597), GTPase-activating protein (Gyp2; UV8b_04168), cAMP-dependent protein kinase regulatory sub-unit (UV8b_04860), and regulator of G protein signaling pathway (UV8b_04229); see Table 1, Figure 9B and Appendix A. These results provide evidence supporting our hypothesis that UvSte50 interacts with the MAPK modules Ste11-Ste7 and UvHog1. Therefore, UvSte50 might govern the fungal pathogenicity of *U. virens* by means of the cAMP and MAPK signaling pathways.

### 3.10. RA Domain, but Not SAM Domain, Is Essential for the Interaction of UvSte50 with UvSte7 and UvRas2

UvSte50 is known to interact with UvSte11 (MEKK) and UvSte7 (MEK) in *U. virens* (Table 1, Figure 9B). As UvSte50 contains SAM and RA domains (Figure 1A), in order to determine the role of the two conserved domains in the interaction of UvSte50 with UvSte11 and UvSte7, we generated the prey constructs of UvSte50-SAM (1–131 aa) and UvSte50-RA (379–499 aa). In yeast two-hybrid assays, no detectable interaction was observed between UvSte50-SAM and UvSte7; however, UvSte50-RA strongly interacted with UvSte7 (Figure 10 and Appendix A), indicating that the RA domain, but not the SAM domain of UvSte50, is involved in the interaction with UvSte7. We also assayed the interaction of UvSte11 with UvSte50-SAM and UvSte50-RA. Both UvSte50-SAM and UvSte50-RA interacted with UvSte11 (Figure 10 and Appendix A), indicating that SAM and RA were both important for the UvSte50–UvSte11 interaction. In addition, no detectable interaction was observed between the middle region of UvSte50 (residues 132–378) and UvSte11 or UvSte7 (Figure 10 and Appendix A). Therefore, the exact UvSte11–interacting site on UvSte50 is not clear.

*S. cerevisiae* Ste50 interacts with Ras1 and Ras2 by the RA domain, implicating the Ras/cAMP signaling pathway [21,22]. We identified a direct interaction between UvSte50 and UvRas2 (Figure 9B) and therefore, the RA domain of UvSte50 can be considered essential for binding to UvRas2 (Figure 10).

## 4. Discussion

In this study, we investigated the role of the adaptor protein UvSte50, a component of the MAPK cascade. We observed that UvSte50 is involved in the regulation of conidiation, virulence, and osmoadaptation stress tolerance in *U. virens*. The vegetative growth of the Δ*Uvste50* mutant was unaffected (Figure 2A), similar to that of *M. oryzae* [26], whereas the Ste50 mutants of *B. cinerea* and *F. graminearum* presented less-developed aerial mycelium and slower radial expansion [25,28]. The function of Ste50 in the regulation of conidia production seems to be conserved, as the Ste50 mutants of *U. virens* (Figure 2B), *M. oryzae* [26], *B. cinerea* [28], and *F. graminearum* [25] all exhibited significant decreases in conidiation. Therefore, different fungi may have recruited conserved signaling pathways for regulation of conidiation.

In *Cryptococcus neoformans*, the function of Ste50 is limited to regulating mating/filamentous growth, and does not participate in the production of virulence factors [45,46]. In contrast, the Ste50 ortholog is essential for pathogenicity in *U. maydis* [27], *M. oryzae* [26], and *F. graminearum* [25]. The *U. virens* hyphae germinated from conidia enter the inside of the glume, initially infects the stamen filaments, and then extends to embrace the entire flower organ [1,2]. Some mutants of *U. virens* lost the ability to infect rice flower organs; for example, the hyphae of the mutant ∆*UvCGBP1* were restricted to the surface of rice spikelets and could not extend into the spikelets [19]. Other mutants of *U. virens* lost the ability to fully utilize the nutrition from rice; for example, the mutant Δ*Uvcom1* could normally infect rice flower organs such as filaments, stigmas, and styles, but the mycelia in the spikelets could not expand to form typical false smut balls [34]. In this study, the mutant Δ*Uvste50* could enter the spikelets and produce observable hyphae on the surface of filaments; however, the hyphae failed to extend to other floral organs, such as stigmas or styles (Figure 3). Therefore, UvSte50 may play an important role in the specific infection of rice filaments in *U. virens*.

UvSte50 was observed to be distributed as spots in the cytoplasm of hyphae of *U. virens* (Figure 9A), different from the Ste50 sub-cellular localization in *M. oryzae* or *F. graminearum*. In *M. oryzae*, *MST50* (an ortholog of Ste50) is expressed at a relatively low level in mycelia and conidia, and the GFP signals were enhanced during appressorium formation and penetration [26]. In *F. graminearum*, FgSte50 is mainly localized on the cell membrane of the hyphae [25]. Ste50 orthologs in different fungal species have different localization patterns, potentially indicating the functional diversity of Ste50 orthologs.

Ste50 in *S. cerevisiae* is involved in cell wall integrity regulation through the Ste11-dependent pathway [47,48], osmoadaptation regulation through the HOG pathway [49,50], and stress-tolerance regulation through interaction with the Ras/cAMP signaling pathway [22]. Meanwhile, in *C. neoformans* and *F. graminearum*, Ste50 is not involved in any of the known stress responses [25,46]. In *U. virens*, Ste11-dependent pathway MAP kinase UvPmk1 mutants showed increased tolerance to hyperosmotic and cell wall stresses, but decreased tolerance to oxidative stress [17]; the HOG pathway MAP kinase UvHog1 mutants had increased sensitivities to hyperosmotic stress and cell wall and membrane stresses, but not oxidative stress [20]; and the Ras/cAMP signaling pathway adenylate cyclase UvAc1 mutants exhibited increased sensitivity to hyperosmotic stress and decreased sensitivity to cell wall stresses CR [41]. Similar to the Δ*Uvhog1* mutant, the Δ*Uvste50* mutant presented increased sensitivities to hyperosmotic stress (via NaCl and sorbitol); however, unlike the Δ*Uvhog1* and Δ*Uvpmk1* mutants, UvSte50 appeared not to be involved in the regulation of cell wall integrity, as the Δ*Uvste50* mutant showed no difference in resistance to SDS, CFW, and CR stress, when compared to the wild-type strain (Figure 6). Furthermore, the Δ*Uvste50* mutant exhibited increased sensitivity to oxidative stress similar to the Δ*Uvpmk1* mutant (Figure 6), but differing from the Δ*Uvhog1* mutant. Deletion of *UvSTE50* had different effects on the expression profile in response to NaCl or H_2_O_2_ treatment. As *UvHOG1* and *UvSSK2* were significantly reduced in the Δ*Uvste50* mutant in the presence of NaCl (Figure 7), *UvSTE50* may regulate response to NaCl treatment through the HOG pathway in *U. virens*. Besides three HOG pathway genes, *UvPMK1*, *UvSTE7*, and *UvSTE11* were also significantly reduced in the Δ*Uvste50* mutant in the presence of H_2_O_2_ (Figure 7); therefore, *UvSTE50* may regulate the response to H_2_O_2_ treatment through the UvPmk1–MAPK and UvHog1–MAPK pathways in *U. virens*. Thus, UvSte50 is involved in regulating stress responses via multiple signaling pathways in *U. virens*.

In *S. cerevisiae*, Ste50 is used as an adaptor protein to couple the Cdc42–Ste20 complex to Ste11 (MEKK) [21]. The association between the homologs of Ste50 and Ste11 has also been reported in *S. pombe* [51], *U. maydis* [27], *F. graminearum* [25], and *M. oryzae* [26]. The Ste50 adaptor protein contains two evolutionarily conserved domains, including a SAM (sterile alpha motif) and a RA (Ras Association) domain [49,50]. SH3 domains have also been identified in the Ste50 orthologs of basidiomycetous fungi, which were required for pathogenicity that typically bind the Ste11 orthologues and Ras, but not the Ste7 (MEK) or Fus3 (MAPK) components, such as Ubc2 (homologous to yeast Ste50) in *U. maydis* [27,52]. The SAM domain is a protein-binding or RNA-binding domain involved in signal transduction and transcriptional regulation [53,54,55,56]. The roles of the SAM domain in Ste50 and Ste11 interactions have been well characterized in *S. cerevisiae* [57,58], *F. graminearum* [25], and *M. oryzae* [26]. We also detected an interaction between UvSte50 and UvSte11 in *U. virens* (Table 1, Figure 9B), where both the SAM and RA domain were required for the UvSte50–UvSte11 interaction (Figure 10). However, the middle region (132–378 aa) between the SAM and RA domain alone is not sufficient for UvSte50 to interact with UvSte11 (Figure 10). Therefore, the exact UvSte11-interacting site on UvSte50 in *U. virens* remains unclear.

In *S. cerevisiae*, Ste50 did not interact with Ste7 in yeast two-hybrid assays [59]. Furthermore, in *U. maydis*, no direct interactions between Ubc2 (Ste50 ortholog) and Ubc4/Fuz7 (Ste7 orthologs) were observed [27]. In contrast, a direct interaction between Ste50 and Ste7 has been observed in *M. oryzae* [26] and *F. graminearum* [25]. Both the SAM and RA domain of Mst50 in *M. oryzae* are dispensable for the Mst50–Mst7 interaction [26], while, in *F. graminearum*, only the RA domain is necessary for the interaction between FgSte50 and FgSte7 [25]. Our yeast two-hybrid assays also suggested that the RA domain, but not SAM domain, is essential for the interaction between UvSte50 and UvSte7 in *U. virens* (Figure 10).

The RA domain of Ste50 in *S. cerevisiae* is required for proper localization of the cargo proteins delivered by Ste50 [60,61], which control HOG pathway activation through interaction with the cytoplasmic single-transmembrane protein Opy2 [55,56,62]. We detected a direct interaction between UvSte50 and UvHog1 in *U. virens* (Table 1, Figure 9B). The *UvHOG1* gene is the MAP kinase of the HOG1 signaling pathway, which regulates hyphal growth, stress responses, and secondary metabolism [20]. The direct interaction between UvSte50 and UvHog1 is a unique phenomenon in *U. virens*. Thus, UvSte50 may regulate pathogenicity through UvHog1–MAPK as well as UvPmk1–MAPK. In addition to its role as an adaptor for Ste11, Ste50 has also been implicated in the Ras/cAMP signaling pathway, potentially due to the presence of the RA domain at the C-terminus [21,22]. Indeed, Ste50 also interacts with Ras1 and Ras2 through the RA domain in *S. cerevisiae* [21] and *M. oryzae* [26]. We identified the direct interaction between UvSte50 and UvRas2 (Figure 9B), and the RA domain of UvSte50 was found to be essential for binding to Ras2 (Figure 10). Our affinity capture–mass spectrometry (MS) assay and yeast two-hybrid data indicated that UvSte50 also interacts with UvAc1 (Adenylate cyclase) and UvCap1 (cyclase-associated protein) in *U. virens* (Table 1, Figure 9B). UvAc1 and UvCap1 are conserved components of the cAMP pathway which are important for conidiation, stress responses, virulence, and regulation of the intracellular cAMP level in *U. virens* [31,41]. The direct interaction between UvSte50 with UvAc1 and UvCap1 indicated that UvSte50 may regulate cAMP/PKA pathways in *U. virens*. Although UvSte50 interacted with UvAc1 and UvCap1, the expression level of *UvAC1* and *UvCAP1* genes were not influenced by the deletion of *UvSTE50* (Appendix A), and the intracellular cAMP level of Δ*Uvste50* showed no difference with the wild-type strain Jt209 (Appendix A). Thus, further studies are needed to explore the role of UvSte50 in the cAMP/PKA pathways and MAPK signaling pathways, in order to better understand the diverse functions of Ste50 orthologs in different fungal systems.

## 5. Conclusions

Ste50 is known to be involved in multiple signaling pathways in the budding model yeast *S. cerevisiae*. Our study functionally characterized the *UvSTE50* gene in *U. virens*. *UvSTE50* plays essential roles in conidiation, stress responses regulation, and pathogenicity. Affinity capture–mass spectrometry analysis and a yeast two-hybrid assay identified the interactions of UvSte50 with Fus3-MAPK pathway cascades, including UvSte11 and UvSte7; the Hog1-MAPK pathway MAP kinase UvHog1; and Ras/cAMP signaling pathway components, including UvRas2, UvAc1 (adenylate cyclase), and UvCap1 (cyclase-associated protein). In conclusion, UvSte50 plays a key role in the specific infection with rice filaments through regulating the MAPK and Ras/cAMP signaling pathways in *U. virens*.

## Figures and Tables

**Figure 1 jof-08-00954-f001:**
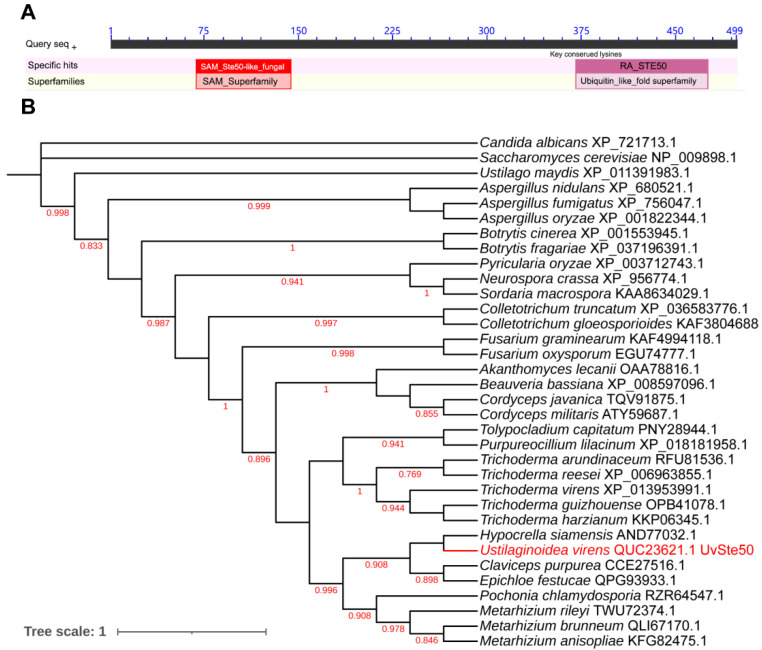
Ste50 protein is conserved among different fungi. (**A**) UvSte50 contains two conserved domains: a sterile alpha motif (SAM) domain (72–129 aa) and a Ras-association (RA) domain (379–466 aa). (**B**) The phylogenetic tree of fungal Ste50 orthologs. All sequences of Ste50 orthologs were downloaded from NCBI and their accession numbers are labeled on the right. The numbers indicate bootstrap values.

**Figure 2 jof-08-00954-f002:**
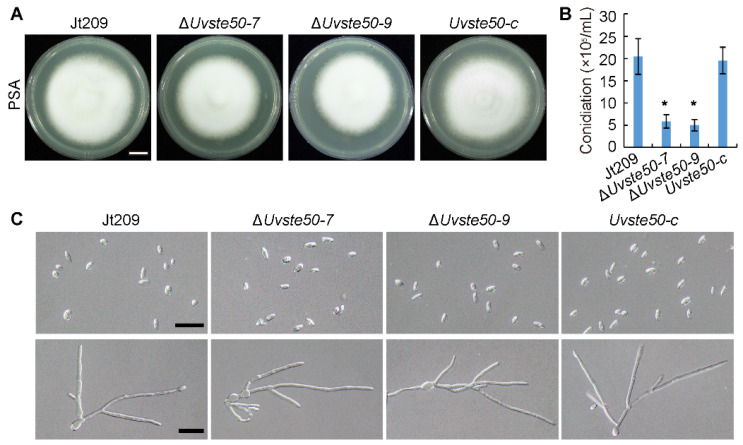
*UvSTE50* affects conidiation in *U. virens*. (**A**) Colony morphology of the wild-type strain Jt209, the Δ*Uvste50* mutant, and the complemented strain *Uvste50-c*. All of the tested strains were incubated on PSA plates for 20 days. Bar, 5 mm. (**B**) Conidial production of Jt209, Δ*Uvste50* and *Uvste50-c* in liquid PSB, incubated for 6 days at 28 °C, and 150 rpm. Error bars represent SD. * indicates significant differences between Δ*Uvste50* and the wild-type/complemented strains, as estimated by Duncan’s new multiple range test (*p* < 0.05). (**C**) Conidial morphology and conidia germination of strains Jt209, Δ*Uvste50*, and *Uvste50-c*. Conidia were harvested from 6-day-old liquid PSB cultures, and conidial germination was observed 20 h post-incubation. Bar, 20 μm.

**Figure 3 jof-08-00954-f003:**
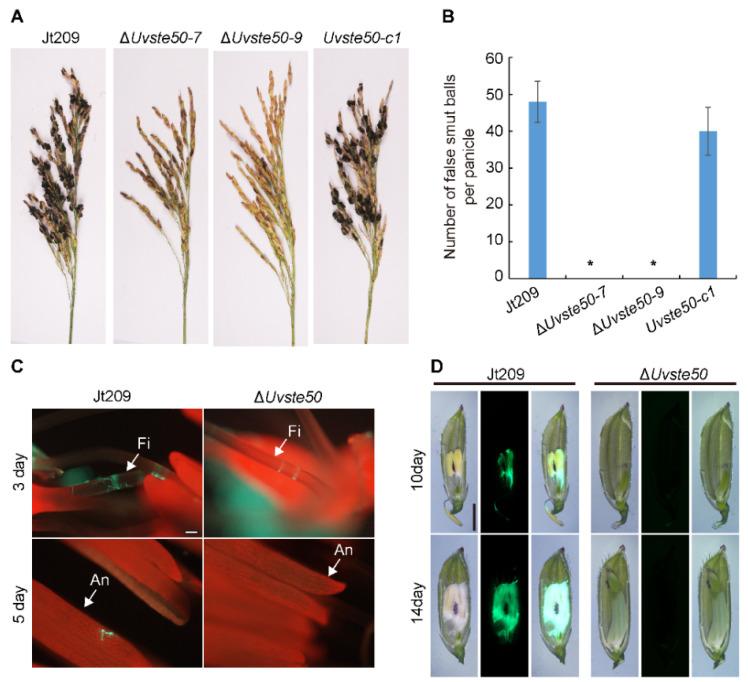
Virulence assay and infection development of Δ*Uvste50* mutant on rice panicles. (**A**) Virulence of the wild-type strain Jt209, the Δ*Uvste50* mutant, and its complemented strain *Uvste50-c* at 30 days post-inoculation (dpi) on susceptible rice cultivar Liangyoupeijiu. (**B**) Average number of false smut balls per panicle. Data were collected from three independent experiments. * represent significant differences between the Δ*Uvste50* mutant and the wild-type/complemented strains, analyzed by Duncan’s new multiple range test (*p* < 0.05). (**C**) Mycelial extension of GFP-tagged Jt209 and Δ*Uvste50* on the inner floral organs at 3 and 5 dpi. Bar, 20 μm. (**D**) Mycelial extension of GFP-tagged Jt209 and Δ*Uvste50* inside the spikelets at 10 and 14 dpi. Bar, 10 μm.

**Figure 4 jof-08-00954-f004:**
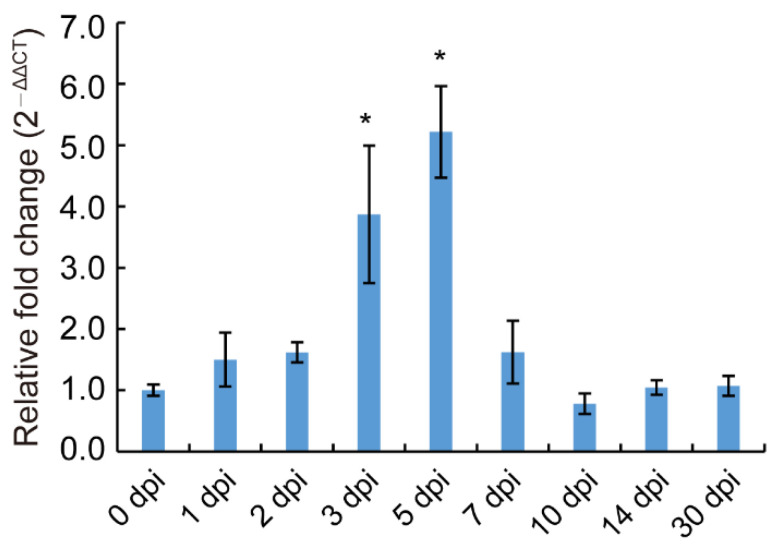
Expression pattern of *UvSTE50* during the infection process of *U. virens*. The expression level of *UvSTE50* relative to *β-TUBULIN* gene at different infection stages on inoculated spikelets (0 to 30 days) were tested by RT-qPCR. Error bars represent SD. * represent significant differences compared with 0 dpi, as estimated by Duncan’s new multiple range test (*p* < 0.05).

**Figure 5 jof-08-00954-f005:**
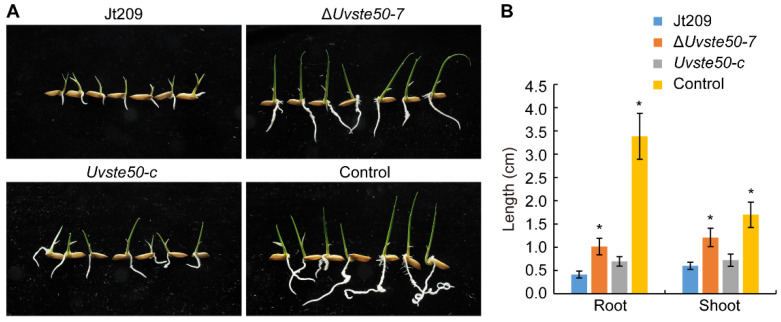
Assays for toxicity of *U. virens* culture filtrates with rice seeds. (**A**) Seeds of rice cultivar Liangyoupeijiu were incubated on sterile filter papers soaked with filtrates of 5-day-old PSB cultures of the wild-type strain Jt209, Δ*Uvste50* mutant, and the complemented strain *Uvste50-c*. Shoot and root growth were examined after incubation at 25 °C for 5 days. (**B**) The statistical analysis of the shoot and root length in (**A**). Data were represented as means ± SD from three independent experiments. * represent significant differences compared with wild-type Jt209 as estimated by Duncan’s new multiple range test (*p* < 0.05).

**Figure 6 jof-08-00954-f006:**
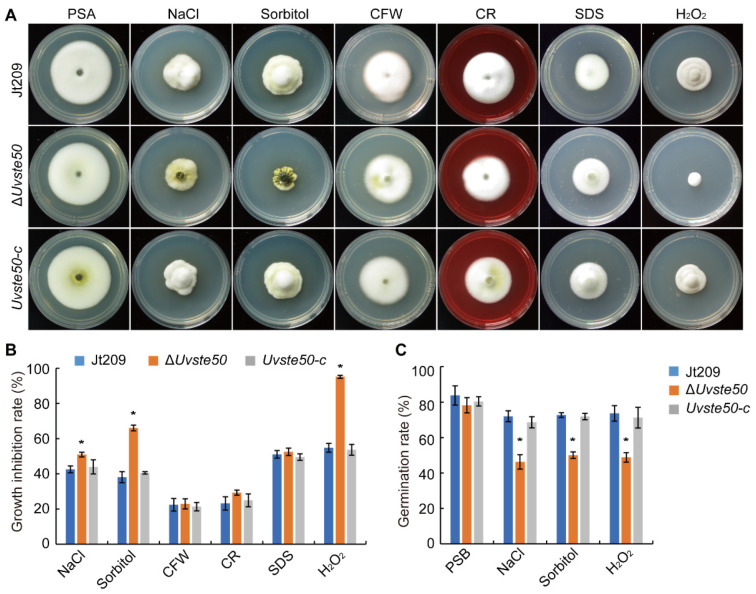
*UvSTE50* regulates stress responses in *U. virens*. (**A**) Mycelial radial growth of the wild-type strain Jt209, Δ*Uvste50* mutant, and the complemented strain *Uvste50-c* on PSA or PSA containing exogenous stress agents including 0.5 M NaCl, 0.6 M sorbitol, 500 μg/mL CFW, 100μg/mL CR, 0.05% SDS, or 0.05% H_2_O_2_. Photographs were obtained after incubation at 28 °C for 20 days in the dark. (**B**) The growth inhibition rate of strains cultured on plates with different stress agents. Measurements of growth inhibition rate were calculated relative to the growth rate of each untreated control. Mean and standard deviation were calculated from three replicates. (**C**) Conidia germination rates were analyzed after treatment without or with 0.3 M NaCl, 0.3 M sorbitol or 0.01% H_2_O_2_ for 20 h. Bar, 20 μm. * represent significant differences between the mutant Δ*Uvste50* and wild-type strain Jt209, as estimated by Duncan’s new multiple range test (*p* < 0.05).

**Figure 7 jof-08-00954-f007:**
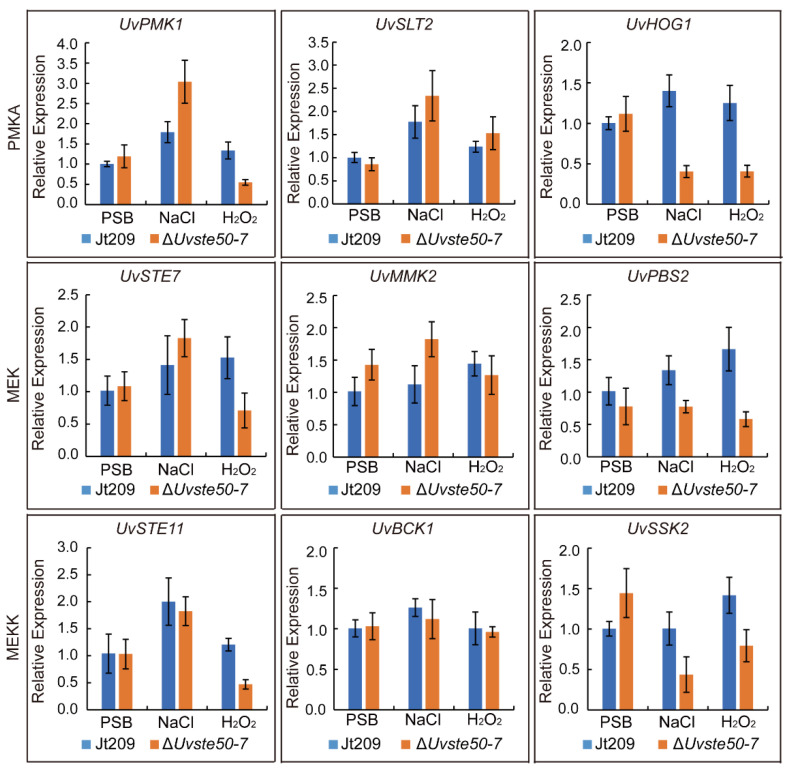
Expression profiles of genes involved in MAPK signaling pathways. RNA samples were isolated from vegetative hyphae of the wild-type strain Jt209 and the Δ*Uvste50* mutant cultured in regular PSB or PSB with 0.5 M NaCl or 0.05% H_2_O_2_. The expression level of each gene in the wild-type strain Jt209 cultured in regular PSB was arbitrarily set to 1.0. Mean and standard deviations were calculated considering the results of three independent replicates.

**Figure 8 jof-08-00954-f008:**
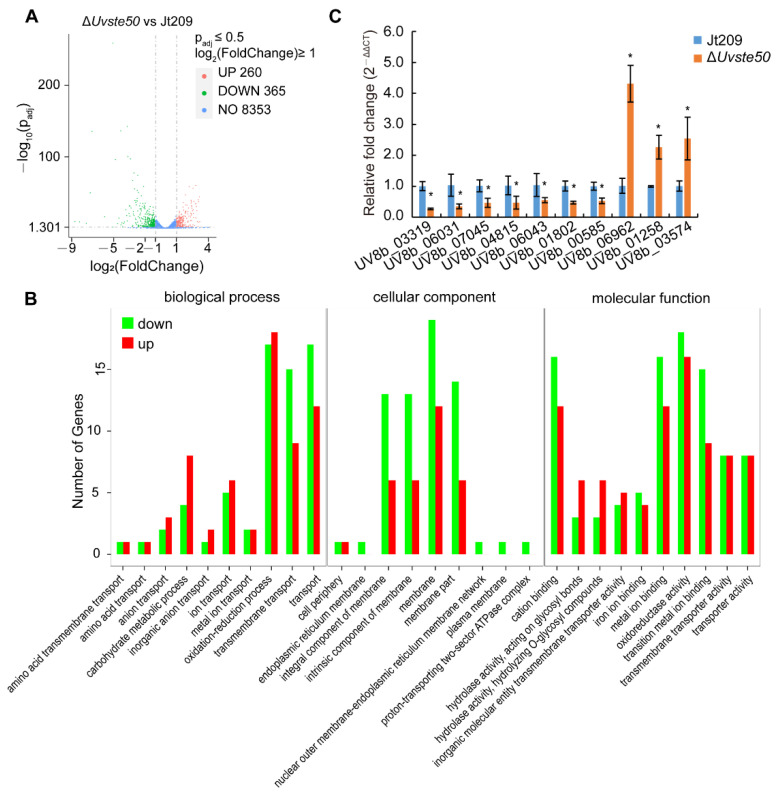
*UvSTE50* affects the transcription of a subset of genes in *U. virens*. (**A**) Volcano map of differentially expressed genes (DEGs). (**B**) Histogram of GO enrichment analysis. (**C**) The expression levels of nine MFS transporter genes and *UvCDTF* (UV8b_03574) in the Δ*Uvste50* mutant. Error bars represent SD. * indicates that the expression level of genes in the Δ*Uvste50* mutant significantly differs from that in the wild-type strain Jt209, as estimated by Duncan’s new multiple range test (*p* < 0.05).

**Figure 9 jof-08-00954-f009:**
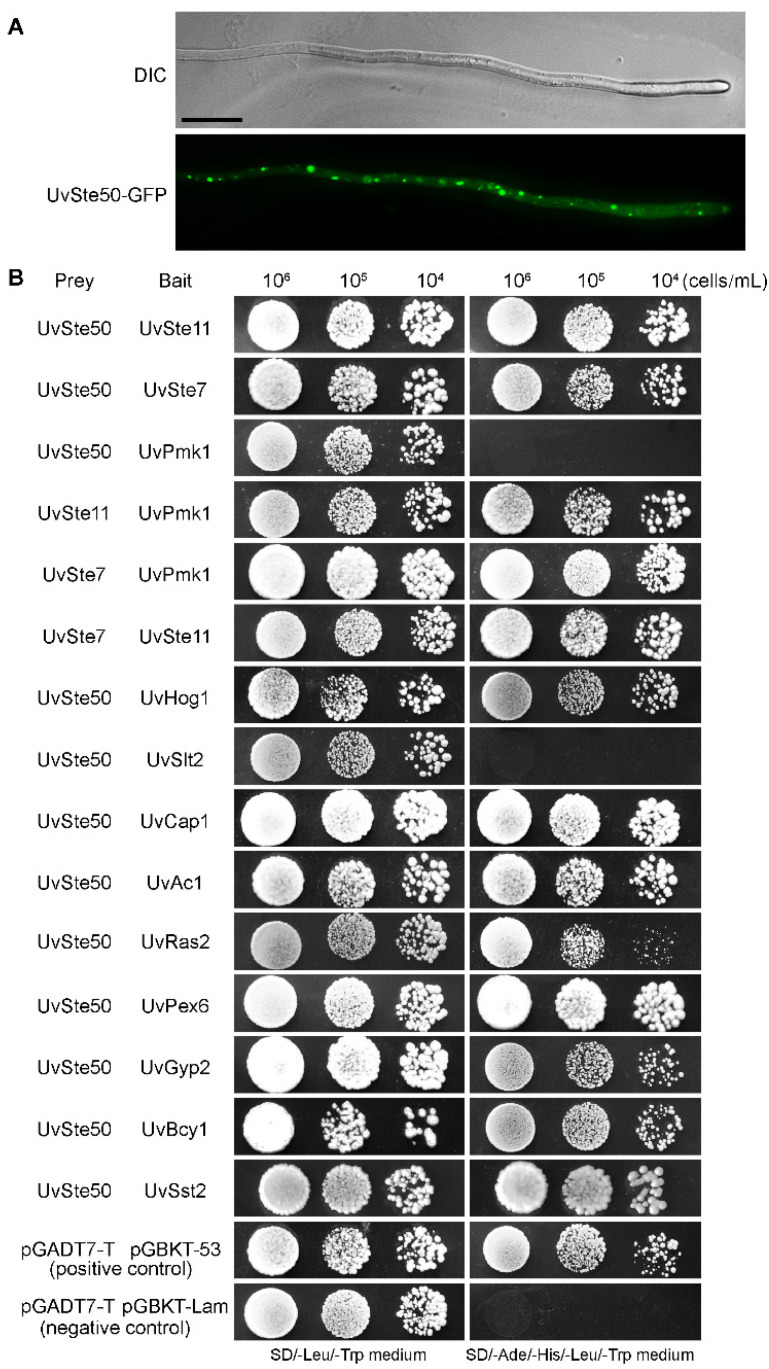
Assays for interactions between UvSte50 and other proteins in *U. virens*. (**A**) Sub-cellular localization of UvSte50 in *U. virens*. Vegetative hyphae of transformant expressing UvSte50-GFP were observed under confocal fluorescence microscopy. Bar, 10 μm. (**B**) A yeast two-hybrid assay was performed to examine the interaction between UvSte50 and other proteins in *U. virens*. Yeast cells (10^4^–10^6^ cells/mL) of transformants containing prey and bait vectors were assayed for growth on SD/-Leu/-Trp and SD/-Ade/-His/-Leu/-Trp medium.

**Figure 10 jof-08-00954-f010:**
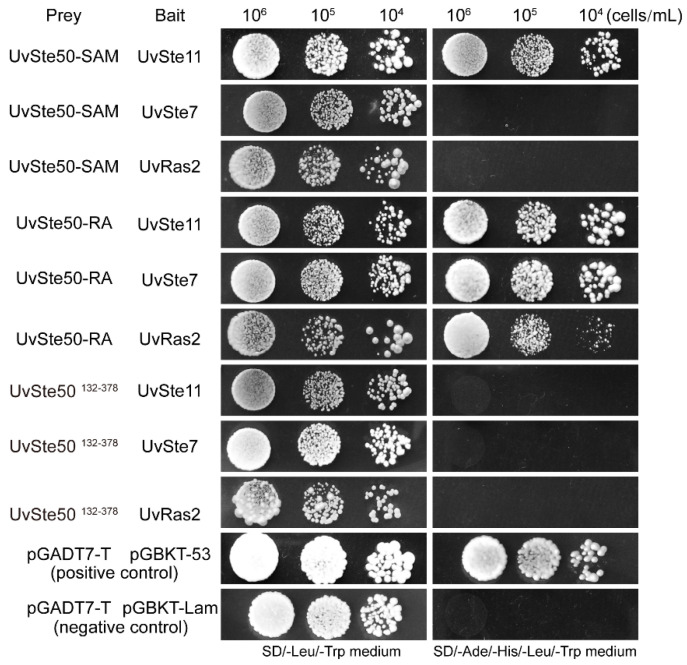
SAM and RA domains play different roles in the interactions of UvSte50 with UvSte11, UvSte7, and UvRas2. UvSte50-SAM (1–131 aa), UvSte50-RA (379–499 aa), and UvSte50^132–378^ (the UvSte50 middle region) constructs were assayed for their interactions with the UvSte11, UvSte7, and UvRas2 bait constructs. The resulting yeast transformants were assayed for growth on SD/-Leu/-Trp and SD/-Ade/-His/-Leu/-Trp medium.

**Table 1 jof-08-00954-t001:** UvSte50-interacting proteins identified by the affinity capture–mass spectrometry assay.

Locus in *U. virens*	Putative Function	Ortholog in *S. cerevisiae*
UV8b_04866	MAP kinase kinase EMK1	Ste7
UV8b_06470	MAP kinase kinase kinase Ste11	Ste11
Uv8b_04241	stress-activated MAP kinase	Hog1
Uv8b_02467	adenylate cyclase	Ac1
Uv8b_00969	putative adenylate cyclase-associated protein	Srv2
UV8b_06597	Peroxisomal biogenesis factor 6	Pex6
UV8b_04168	GTPase activating protein (Gyp2)	Gyp2
UV8b_04860	cAMP-dependent protein kinase regulatory subunit	Bcy1
UV8b_04229	regulator of G protein signaling pathway	Sst2

## Data Availability

RNA-seq data were submitted to NCBI BioProject (ID: PRJNA878941).

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
