# Peer review of "The Adaptor Protein UvSte50 Governs Fungal Pathogenicity of Ustilaginoidea virens via the MAPK Signaling Pathway"

_jof, 2022, doi:10.3390/jof8090954_

Round 1

Reviewer 1 Report

Refer to the attached file.

Reviewer 2 Report

This study describes the identification, isolation and characterization of a Ste50 orthologue from Ustilaginoidea virens, the ascomycete rice false smut fungus. UvSte50 shares homology at the amino acid level with the Ste50 orthologues in ascomycete genetic models, e.g. S. cerevisiae, Sordaria, Neurospora, as well as both human, insect, and plant pathogens (e.g., C. albicans, Metarhizium, Fusarium oxysporum and graminearum, and Botrytis), as well as homologues that include the basidiomycete-specific versions found in Cryptococcus neoformans and Ustilago maydis (i.e., Ubc2). The authors show that, like the other well-characterized Ste50 orthologues, UvSte50 participates as an adaptor in one or more MAPK cascades, especially associated with osmoregulation and mating/filamentation. Also, UvSte50 is found to be required for full pathogenicity. Overall, the investigation is thorough and, for the most part the ideas are conveyed well. There are some issues with language usage, typos, and sufficient description of methods. These are indicated below.

Minor issues:

Abstract, Page 1, Line 19: Change to “We determined that UvSte50 directly…”; Line 25-26: Change to “Taken together…”

Introduction, Page 1, Lines 35-36: Change to ”Besides causing yield losses…food security, since its mycotoxins…nervous systems of animals [4]”

Page 2, Line 63: Change to “pathogenicity was not mentioned in this article reporting its characterization.”

Line 83 (and elsewhere): Need to clarify that the basidiomycete-specific adaptors, e.g., Ubc2 in U. maydis, contain C-terminal SH3 domains required for pathogenicity that typically bind the Ste11 orthologues and Ras, but not the Ste7 (MAPKK) or FUs3 (MAPK) components.

Materials and Methods, Page 3, Line 123: Change to “…and the colony diameters were measured.”; line 127 and throughout: Change to “1 x 106”; line 131: Italicize U. virens; line 145: Change to “hyperosmotic”

Page 4, Line 156, line 160-163: What were the growth conditions for cells where the RNAs were extracted?; line 160: Change to: “…mutant were extracted…”

Page 5, line 198-199: Change to  “…supernatant were analyzed… Then, the elution proteins were analyzed by mass spectrometry…”

Results, Page 5, line 210: What is the significance of the finding that UvSte50 most closely matched the orthologue from Hypocrella siamensis?

Fig. 1. How was the phylogenetic tree generated? How many bootstraps were carried out?

Page 6, Lines 221-221: It is indicated that wild type and DelUvSte50 mutant grew similarly. On what medium? The text says PSB, but Fig. 2 legend says PSA; which is correct?

Line 252: Change to “and no false smut balls were visible…”

Page 8, Section 3.4. Where are the statistical measures for significance for these data?

line 291: Change to “Rice shoots also grew longer…”

Fig. 5. What does CK mean? In 5B, what is the statistical analysis? Where are the measures of significant differences, if any? How were they done?

Page 9, line 309: Change to NaCl”: line 321: Change to “and the wild-type…”; Fig. 6 legend: How was growth inhibition rate measured?

Page 10, line 345: Change to “We tested the …”; line 354: Change to “other components…”

Page 12, Figure 8 and all Y2H work: Were “swaps” done to assure that there was no bias due to the vector (bait vs. prey) for the respective interaction tests? For example, as UvSte50 ever expressed from Bait and UvSte11 from Prey to assure that the reaction as of similar strength? Thismight also have significance, especially for examples where interaction was not found, but also to make certain interactions observed were bona fide.

Page 13, lines 405: Change to: “To test this, we conducted”; lines 406-408: Change to “UvSte50 interacted with…Although there was no interaction…UvSte11 interacted with…”; line 412: Change to “capture mass-spectrometry assay included…”

Lines 427-436: Change to: “UvSte50-RA strongly interacted…UvSte50-RA interacted with Ste11…SAM ad RA were both important…no detectable interaction as observed between. We identified direct iteraction…”

Discussion, Page 14, Line 447: Change to We observed…”; line 450: Change to “whereas, the Ste50 mutants…”; line 454: Change to “Different fungi…”; line 458: Change to In contrast, the Ste50…” ; Line 461-463: This sentence does not make sense as written.

Page 15, line 489-490: Change to “Similar to the DUvhog1 mutant…however, unlike the …”; line 495: Change to “Thus, UvSte30 is involved…”

Page 16, lines 434-439: Change to ‘…are conserved components…may regulate cAMP-PKA pathways…Although UvSte50 interacted with…”

Round 2

Reviewer 1 Report

Now, the revised version is acceptable for publication.